Review

# CXCR4: from B-cell development to B cell–mediated diseases

Stéphane Giorgiutti[1,2,3], Julien Rottura[2], Anne-Sophie Korganow[1,2,3], Vincent Gies[1,2,4]

Chemokine receptors are members of the G protein–coupled receptor superfamily. The C-X-C chemokine receptor type 4 (CXCR4), one of the most studied chemokine receptors, is widely expressed in hematopoietic and immune cell populations. It is involved in leukocyte trafficking in lymphoid organs and inflammatory sites through its interaction with its natural ligand CXCL12. CXCR4 assumes a pivotal role in B-cell development, ranging from early progenitors to the differentiation of antibody-secreting cells. This review emphasizes the significance of CXCR4 across the various stages of B-cell development, including central tolerance, and delves into the association between CXCR4 and B cell–mediated disorders, from immunodeficiencies such as WHIM (warts, hypogammaglobulinemia, infections, and myelokathexis) syndrome to autoimmune diseases such as systemic lupus erythematosus. The potential of CXCR4 as a therapeutic target is discussed, especially through the identification of novel molecules capable of modulating specific pockets of the CXCR4 molecule. These insights provide a basis for innovative therapeutic approaches in the field.

## Introduction

Chemokine receptors are heptahelical surface molecules, typically coupled to G proteins, which respond to a large family of small secreted proteins. They play a pivotal role in cell migration, most notably leukocytes (Hughes & Nibbs, 2018). Among these receptors, the C-X-C chemokine receptor type 4 (CXCR4), alternatively known as a cluster of differentiation 184 (CD184) or fusin (Murphy et al, 2000), is a widely expressed chemokine receptor present in various tissues, including the brain, heart, and kidney. Its predominant expression is observed in hematopoietic and immune cells (Zou et al, 1998; Pawig et al, 2015). Along with its natural ligand CXCL12, also named SDF-1 (stromal cell–derived factor 1), CXCR4 plays important roles in leukocyte trafficking in lymphoid organs and

inflammatory sites (Janssens et al, 2018). It has critical physiological roles from the earliest stages of embryonic development to the extent that the loss of CXCR4 function leads to lethality during the perinatal period in mice (Ma et al, 1998). In pathological contexts, CXCR4 is implicated in both cancer development and progression, as well as viral infections, particularly in the case of HIV-1, where CXCR4 serves as the coreceptor for CD4 on T cells during infection (Deng et al, 1996; Guo et al, 2016).

The significant role of CXCR4 in the development, differentiation, and homeostasis of B cells is less well known but is increasingly being demonstrated. Long-standing observations have established that mice lacking CXCL12 exhibit compromised maturation of B cells, indicating the key contribution of CXCR4 (Nagasawa et al, 1996). In humans, gain-of-function (GOF) mutations in *CXCR4* are associated with lymphopenia and hypogammaglobulinemia in the so-called WHIM (warts, hypogammaglobulinemia, infections, and myelokathexis) syndrome (Hernandez et al, 2003).

In this review, we provide an overview that emphasizes key structural and signaling aspects of CXCR4 and underscores the significance of this receptor in various stages of B-cell development, spanning from hematopoietic stem cells (HSCs) to the emergence of antibody-producing cells. In addition, we explore the potential involvement of this receptor in pathological processes associated with non-malignant B-cell diseases.

## CXCR4: structure and signaling

### CXCR4 structure

CXCR4 is a member of the G protein–coupled receptor family, characterized by seven alpha helices that traverse the cellular membrane. Situated on chromosome 2, the *CXCR4* gene encodes a 352–amino acid protein (Caruz et al, 1998). The receptor's sequence is highly conserved between species, showing a sequence homology of ~91% between mice and humans, confirming its indispensable role in species evolution (Heesen et al, 1996; Doranz et al, 1999; Wu et al, 2010).

Within the extracellular segment of CXCR4, two disulfide bonds establish a connection between the N-terminus and the second

---

[1]Department of Clinical Immunology and Internal Medicine, National Reference Center for Systemic Autoimmune Diseases (CNR RESO), Tertiary Center for Primary Immunodeficiency, Strasbourg University Hospital, Strasbourg, France   [2]INSERM UMR - S1109, Institut thématique interdisciplinaire (ITI) de Médecine de Précision de Strasbourg, Transplantex NG, Fédération Hospitalo-Universitaire OMICARE, Fédération de Médecine Translationnelle de Strasbourg (FMTS), Strasbourg, France   [3]Faculty of Medicine, Université de Strasbourg, Strasbourg, France   [4]Faculty of Pharmacy, Université de Strasbourg, Illkirch, France

Correspondence: Stephane.Giorgiutti@chru-strasbourg.fr

extracellular loop, forming the entrance of a ligand-binding pocket (Wu et al, 2010). The negative charges present in this pocket further facilitate the binding of CXCL12 (Qin et al, 2015). The crystallographic structure reveals a division of the binding pocket into a major and a minor subpocket (Scholten et al, 2012; Qin et al, 2015). The major subpocket is delineated by the side chains originating from alpha helices III, IV, V, VI, and VII, whereas the minor subpocket is defined by the side chains arising from alpha helices I, II, III, and VII (Scholten et al, 2012). The precise interactions of CXCL12 with these two subpockets remain currently unclear (Qin et al, 2015; Stephens et al, 2020).

The intracellular side of CXCR4, corresponding to three intra-cellular loops and the C-terminal domain, is structurally more conserved between species (Heesen et al, 1996; Wu et al, 2010). The C-terminal tail is rich in serine/threonine amino acids, constituting targets for phosphorylation events central in CXCR4's downstream signaling cascades and desensitization mechanisms (Busillo et al, 2010; Cronshaw et al, 2010). Remarkably, mutations truncating this C-terminal tail have been linked to the acquisition of enhanced CXCR4 function, a hallmark of WHIM syndrome (see paragraph "*WHIM syndrome*") (Hernandez et al, 2003).

### *CXCR4 signaling*
CXCL12 binding to CXCR4 triggers a conformational change in CXCR4 that activates the heterotrimeric G protein. This activation leads to the exchange of GDP with GTP on the alpha subunit, and disso-ciation of G$\alpha$ subunit from the G$\beta$/G$\gamma$ subunit (Oldham & Hamm, 2008). G$\alpha$ inhibits cyclic AMP production by adenylate cyclase but can activate the Ras/ERK1/2 pathway. G$\beta$/G$\gamma$ dimers trigger in-tracellular calcium mobilization through phospholipase C-$\beta$ (PLC-$\beta$), and activate phosphoinositide 3-kinase (PI3K)/AKT pathways (Scala, 2015). These intracellular signaling pathways induce various cellular responses, such as migration, chemotaxis, proliferation, and survival (Scala, 2015).

A desensitization process, initiated by the binding of CXCL12 to the receptor, involves the $\beta$-arrestin family of proteins. Indeed, activation of CXCR4 allows the recruitment of receptor kinases coupled to G protein kinases (i.e., GRK2, GRK3, GRK6). GRKs phosphorylate the serine–threonine residues of the C-terminal tail, leading to the recruitment of $\beta$-arrestins (Balabanian et al, 2008; Busillo et al, 2010). $\beta$-arrestins interact with clathrin to internalize the receptor and uncouple it from the heterotrimeric G protein (Claing et al, 2002). CXCR4 can then be recycled back to the plasma membrane or be targeted for lysosomal degradation by the E3 ubiquitin ligase atrophin-interacting protein 4 in a $\beta$-arrestin–dependent manner (Bhandari et al, 2007). The $\beta$-arrestin role seems not limited to CXCR4 signaling desensiti-zation, and may have a signaling role of its own, in particular, through ERK pathway activation (Busillo et al, 2010; Jean-Charles et al, 2017).

The dynamic spatial arrangement of the receptor at the cell membrane also plays a crucial role in its functions, although this aspect remains less understood (Martínez-Muñoz et al, 2018). G protein–coupled receptors are organized as monomers, homo- or heterodimers, or even oligomers. CXCR4 can heterodimerize with ACKR3 (also known as CXCR7), an atypical chemokine receptor uncoupled to G proteins. ACKR3 is thought to be a CXCL12 scavenger receptor, and heterodimerization may modulate CXCR4-mediated responses (Sierro et al, 2007; Levoye et al, 2009).

## CXCR4 in B-cell development and functions

CXCR4 is expressed on all B-cell subsets during B-cell development from HSCs to antibody-secreting cells. However, the level of the expression of the receptor varies over time, allowing B-lineage cells to progress in their development and maturation (Aiuti et al, 1997; Honczarenko et al, 1999; Palmesino et al, 2006) (Fig 1).

### *Central development: from HSCs to immature B cells*
CXCR4 is already involved at a very early stage of hematopoiesis in the BM, maintaining a pool of quiescent HSCs. The CXCL12/CXCR4 axis, throughout contacts with the so-called CXCL12-abundant re-ticular (CAR) cells, maintains retention of HSCs within specialized niches (Sugiyama et al, 2006; Tzeng et al, 2011). Indeed, the induced deletion of CXCR4 in adult mice leads to a reduction in the number of HSCs within the BM. HSCs, in the absence of CXCR4, demonstrate heightened vulnerability to myelosuppressive stress triggered by 5-fluorouracil and exit the G0 quiescent phase (Sugiyama et al, 2006). In accordance, the in vitro addition of CXCL12 inhibits the entry of murine HSCs into the cell cycle in a dose-dependent manner, confirming the central role of CXCR4 in governing HSC proliferation (Nie et al, 2008). Depletion of CAR cells in mice results in a reduced number of HSCs and an up-regulation of early myeloid selector genes, resembling the phenotype observed in wild-type HSCs cultured without a niche (Omatsu et al, 2010). Conversely, aberrant CXCR4 signaling is also deleterious to generate early lymphoid progenitors, as the increased quiescence of short-term HSCs ob-served in CXCR4 GOF $Cxcr4^{+/1013}$ mice impairs transition to multi-potent progenitors and to the common lymphoid progenitor (CLP) (Freitas et al, 2017). Hence, both CAR cells and a fine regulation of CXCR4 signaling are essential for the generation of lymphoid progenitors and for the maintenance of HSCs in an undifferentiated state (Omatsu et al, 2010).

Furthermore, CXCR4 enables CLP to position in the vicinity of IL7$^+$CXCL12$^+$ stromal cells, allowing the commitment of CLP in the B-lineage (Tokoyoda et al, 2004; Cordeiro Gomes et al, 2016; Kaiser et al, 2023). Subsequently, pro-B cells in contact with the IL-7–expressing cells proliferate, while remaining anchored within these niches (Tokoyoda et al, 2004) (Fig 1). During this stage, the IL-7 signaling pathway sustains the expression of CXCR4 and the ad-hesion protein FAK augmenting adherence to the stromal envi-ronment (Clark et al, 2014; Fistonich et al, 2018). VDJ recombination ensues, leading to the formation of a functional immunoglobulin heavy chain (Igμ), constitutive of the pre-BCR (Clark et al, 2014). Observations from high-power field confocal microscopy of mouse BM have revealed that small pre-B cells, which cease proliferating, tend to localize in proximity to stromal cells exhibiting low levels of IL-7 but high levels of CXCL12 (Mandal et al, 2019). This strategic localization is attributed to the effect of the pre-BCR, which induces an up-regulation of CXCR4 expression through IRF4, concurrently down-regulating the expression of adhesion factors (Mandal et al, 2019). This orchestrated response enables pre-B cells to disengage from IL-7–rich niches within the BM (Johnson et al, 2008; Fistonich et al, 2018). Between pre-B cells and immature/mature B cells, a

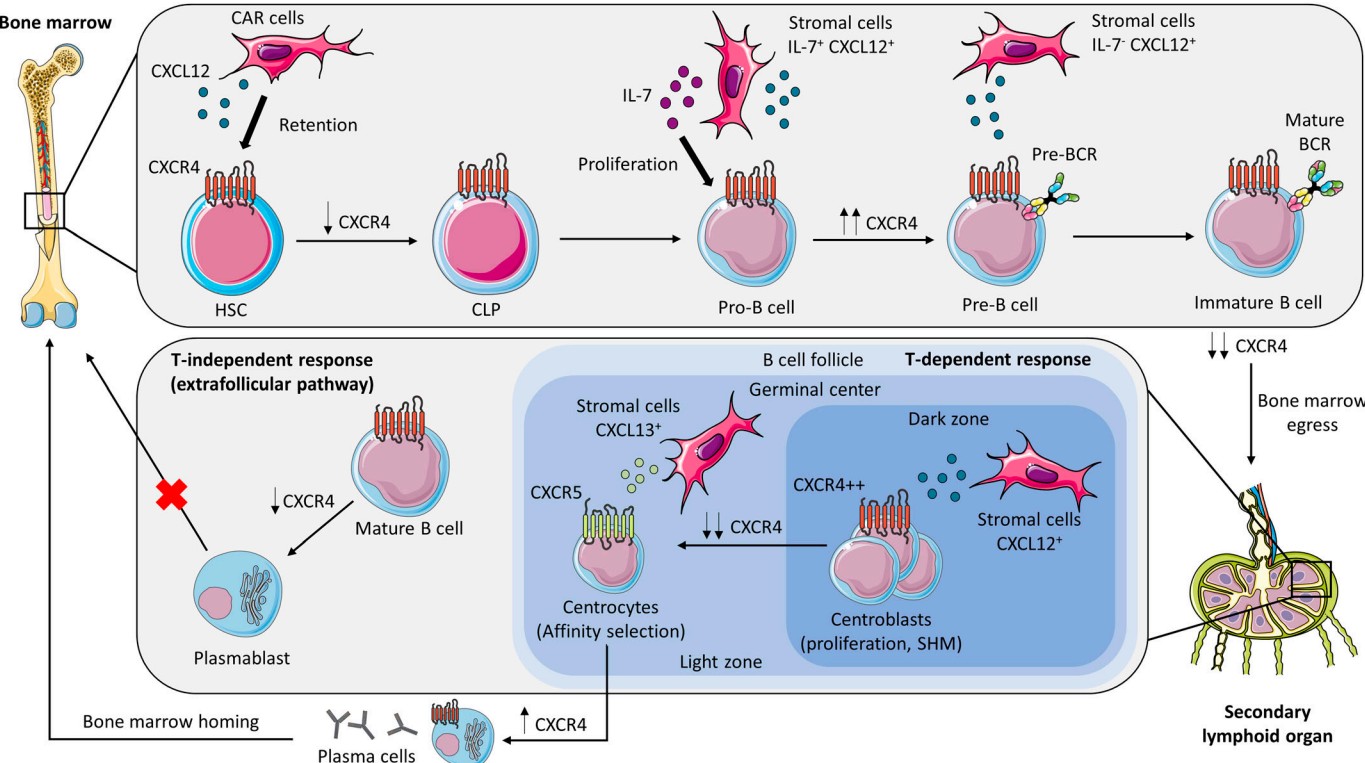

**Figure 1. CXCR4 and B-cell development.**
CXCR4 is involved in maintaining the HSC pool throughout contacts with CXCL12-abundant reticular cells in the BM. The level of CXCR4 expression varies during B-cell differentiation. Pro-B cells undergo intensive proliferation in the vicinity of IL-7+ CXCL12+ stromal cells. An up-regulation of CXCR4 expression characterizes pre-B-cell stage, with disengagement from IL-7–rich niches. Then, the low level of CXCR4 expression in immature B cells induces BM egress to secondary lymphoid organs (see paragraph "*Central development: from hematopoietic stem cells to immature B cells*"). During the T-dependent response in the germinal center, centroblasts in the dark zone, abundant in CXCL12-expressing stromal cells, strongly express CXCR4, and undergo proliferation and somatic hypermutation. Centrocytes down-regulate CXCR4 to enter the light zone through the CXCL13/CXCR5 axis. Plasma cells emerging from the germinal center re-express CXCR4 and accumulate in the BM, where some of them mature into long-lived plasma cells. Regulation of CXCR4 signaling is also required to limit extrafollicular response and avoid plasmablast accumulation in the BM (see paragraph "*From mature B cells to antibodies response*"). BCR, B-cell receptor; CAR, CXCL12-abundant reticular cells; CLP, common lymphoid progenitor; HSC, hematopoietic stem cell; SHM, somatic hypermutation. The figure was partly generated using Servier Medical Art, provided by Servier, licensed under a Creative Commons Attribution 3.0 Unported license.

twofold down-regulation of CXCR4 leads to a decreased response to CXCL12 in mice and allows the egress of these cells from the BM (Honczarenko et al, 1999; Beck et al, 2014) (Fig 1). Immature and mature B cells increase simultaneously CCR7 expression favoring the formation of CXCR4-CCR7 heterodimers, which may also impair CXCR4 signaling and induce BM exit (Mcheik et al, 2019).

Discerning the direct and indirect effects of CXCR4 signaling in vivo poses a considerable challenge. Numerous data from studies conducted in artificial BM progenitor culture systems indicate that CXCR4 exerts effects beyond mere chemotaxis. In vitro, the expression of pre-BCR and evasion of IL-7 only result in minor effects on B-cell developmental transcriptional and epigenetic programs without CXCL12 (Mandal et al, 2019). B220+IgM− progenitors from WT mice, cultivated with a reduced IL-7 dose in the presence of CXCL12, have a transcriptomic profile of cell cycle repression and tend toward differentiation programs involving *Irf4*, *Irf8*, and *Ikzf3* (encoding the transcription factor Aiolos) in an ERK-dependent manner (Mandal et al, 2019). Consistently, ATAC-seq experiments reveal that CXCR4 opens binding sites for transcription factors involved in late B lymphopoiesis (e.g., FOXO1, E2A) while preventing the binding of factors such as MYC and STAT5, implicated in earlier processes (Mandal et al, 2019). CXCR4 is also involved in light-chain recombination in small pre-B cells, favoring *Rag* expression and *Igk* transcription (Mandal et al, 2019; McLean & Mandal, 2020).

In addition to the pivotal role of CXCR4 in B-cell ontogeny, recent data highlight its involvement in central B-cell tolerance. In the 3–83Igi,H-2b mb1Cre mouse model, autoreactive immature B cells, characterized by high avidity for the self-antigen MHC-I H-2Kb, have a 1.5-fold increase in CXCR4 expression compared with non-autoreactive cells, preventing them from migrating to the periphery (Greaves et al, 2019; Pelanda et al, 2022). This CXCR4 higher expression in immature autoreactive B cells is also observed in a human immune system humanized mouse (HIS hu-mice), in which all mouse cells express a self-antigen that reacts with developing human Igκ+ B cells (Alves da Costa et al, 2021). In this model, autoreactive immature B cells show a higher migration potential in response to CXCL12 than non–self-reactive immature B cells. Furthermore, treatment of these mice with a CXCR4 antagonist (AMD3100) results in a twofold increase in the frequency of

autoreactive B cells in the spleen after 48 h, confirming that CXCR4 plays functionally a key role in the retention of autoreactive B cells in the BM (Alves da Costa et al, 2021).

More recently, Okoreeh et al suggested, with in vitro culture of B220⁺IgM⁻ progenitors, the direct requirement for CXCR4 signaling in orchestrating receptor editing and Igλ recombination (Okoreeh et al, 2022). Indeed, CXCR4 deficiency impairs Igλ+ B-cell development, as CXCR4 signaling promotes chromatin accessibility of the Igλ locus and modulates accessibility at binding motifs for transcription factors critical in receptor editing (Okoreeh et al, 2022).

Altogether, CXCR4 not only plays a role in B-cell development in BM but also contributes to central B-cell tolerance mechanisms.

### From mature B cells to antibody response

In the periphery, CXCR4 function on mature naive B cells requires the expression of the IgD-BCR (Becker et al, 2017). In mice, the absence of IgD results in the abolition of CXCL12-induced migration, underlining the pivotal role of IgD in this migratory process. CXCR4 signaling further activates the ERK and AKT pathways in an IgD-dependent manner, and intriguingly, CD19 emerges as a potential intermediary between CXCR4 and IgD-BCR. This is supported by the restoration of CXCR4 signaling upon in vitro stimulation of CD19 in IgD-deficient B cells, indicating a functional connection between CXCR4, CD19, and IgD-BCR during B-cell activation (Becker et al, 2017).

Furthermore, CXCR4 is essential for the T-dependent immune response notably through its role in the mobilization of centroblasts in the germinal center (GC) of secondary lymphoid organs. Centroblasts located in the dark zone of the GC, which contains a network of CXCL12-expressing reticular cells, exhibit strong CXCR4 expression in both mice and humans (Allen et al, 2004; Caron et al, 2009; Rodda et al, 2015). In contrast, centrocytes down-regulate CXCR4, enabling them to leave the dark zone and to enter the light zone directed by the CXCL13/CXCR5 axis (Allen et al, 2004, 2007; Caron et al, 2009; Victora et al, 2010) (Fig 1). Multiple mechanisms may contribute to the differential CXCR4 expression between the light and dark zones. In the light zone, follicular dendritic cells express CD161, which is the binding partner for the C-type lectin-like receptor LLT1 found on the membranes of GC B cells. When human tonsillar GC B cells are exposed to recombinant human CD161 in culture, a decrease in CXCR4 expression is observed, indicating that CD161/LLT1 interactions could potentially trigger the phenotypic transition from the dark to the light zone (Llibre et al, 2016). Another potential mechanism for explaining this transition comes from murine data, which demonstrate that IL-21 produced by follicular helper T cells, along with an increase in CD63 expression, facilitates internalization and may contribute to the down-regulation of CXCR4 in centrocytes (Yoshida et al, 2011). As a result, altered CXCR4 expression could have an impact on the homeostasis of the GC. In accordance with this, altered lymphoid follicle architecture has been reported in CXCR4 GOF Cxcr4⁺/¹⁰¹³ mice and in two histopathological reports of lymph nodes from WHIM patients, but the origin of these abnormalities is difficult to interpret in the context of WHIM-associated lymphopenia (Zuelzer, 1964; Mentzer et al, 1977; Balabanian et al, 2012).

Besides the GC, CXCR4 regulation is also important to control extrafollicular response, that is, the T-independent immune response, which serves as the primary line of defense during infections. In CXCR4 GOF Cxcr4⁺/¹⁰¹³ mice, the defect of CXCR4 desensitization leads to an exacerbated extrafollicular B-cell response, marked by an increase in IgM⁺ splenic plasmablasts and an elevation in total IgM serum titers after T-independent antigen immunization (Alouche et al, 2021). Accordingly, in vitro plasmablast differentiation, with TLR4 ligands, of splenic B cells from Cxcr4⁺/¹⁰¹³ mice is increased compared with controls (Alouche et al, 2021). Mechanistically, the persistence of CXCR4 signaling favors B-cell entry into the cell cycle through the mTORC1 pathway and promotes differentiation into extrafollicular plasmablasts (Alouche et al, 2021). Furthermore, Cxcr4⁺/¹⁰¹³ mice exhibit a fivefold increase in BM plasmablast cells compared with WT mice 3 d after immunization (Alouche et al, 2021). Interestingly, this finding is corroborated in humans, as WHIM patients display a significant increase in CD19⁺CD138⁻ plasmablasts in the BM compared with healthy donors (Alouche et al, 2021). Regulation of CXCR4 expression is therefore necessary to contain the extrafollicular response and the medullary tropism of plasmablasts.

Antigen-specific antibody-producing cells that emerge from the GC also re-express CXCR4, enabling them to accumulate within the BM, where some mature into long-lived plasma cells (Hargreaves et al, 2001; Hauser et al, 2002) (Fig 1). Immunohistological analysis of WT mouse BMs confirms that almost all plasma cells cluster in niches with cells expressing CXCL12 (Tokoyoda et al, 2004). This localization is necessary for plasma cells to benefit from a favorable microenvironment, in particular, to be exposed to survival factors such as APRIL (Cassese et al, 2003; Belnoue et al, 2012). A defect in CXCR4 expression in plasma cells results in their accumulation in the spleen and peripheral blood in mice (Hargreaves et al, 2001; Nie et al, 2004; Tokoyoda et al, 2004), and through time-lapse intravital tibial imaging, Benet et al demonstrate that blocking CXCR4 with an antagonist, AMD3100, inhibits intra-BM plasma cell dynamics (Benet et al, 2021). Paradoxically, in CXCR4 GOF mice, GC-derived antigen-specific plasma cells fail to accumulate in the BM and to induce a long-term immune response (Biajoux et al, 2016). This highlights our incomplete understanding of plasma cell homing and survival, both within and outside the BM environment. However, this counterintuitive finding could be explained by the exacerbated extrafollicular reaction in this mouse model, leading to the accumulation of low-affinity extrafollicular plasmablasts, which competes with antibody-producing cells from the GC reaction for the BM niches (Alouche et al, 2021). This is consistent with a defect of durable humoral T-dependent immune response (i.e., after vaccination) observed in some WHIM patients (Gulino et al, 2004; Handisurya et al, 2010).

Altogether, a fine regulation of CXCR4 signaling is necessary to obtain a specific and durable humoral response. The apparent contradiction between its roles in plasma cell differentiation and HSC quiescence underscores CXCR4's context-dependent functions across cell types.

## B cell–mediated disease and CXCR4

The activation of the CXCR4/CXCL12 axis is a frequent event in a wide spectrum of pathological conditions (Berger et al, 1999; Balkwill, 2004; Teicher & Fricker, 2010). Given its pivotal involvement in B-cell

homeostasis, and beyond its extensively documented role in B-cell lymphoproliferative disorders (Du et al, 2019; Kaiser et al, 2021), CXCR4 can play a contributory role, if not a central one, in non-malignant B-cell disorders, ranging from immunodeficiencies to autoimmune diseases.

### WHIM syndrome

WHIM (warts, hypogammaglobulinemia, infections, and myeloka-thexis) syndrome is an autosomal dominant primary immunode-ficiency caused by GOF mutations of CXCR4 (Hernandez et al, 2003). The pathological variants affect the C-terminal region and lead to defective internalization of CXCR4 (Geier et al, 2022). In a 66-patient cohort, 92% of the patients developed infections including otitis, pneumonia, and human papillomavirus–related manifestations (Geier et al, 2022). 88% of patients suffer from lymphopenia in-volving both T and B cells, and 65% have hypogammaglobulinemia (Geier et al, 2022). Circulating B-cell immunophenotyping shows a reduction in both unswitched and switched circulating memory B cells (Gulino et al, 2004; Mc Guire et al, 2010). Circulating B cells from WHIM patients are more prone to apoptosis and to sponta-neous up-regulation of the activation marker CD69 compared with healthy donors (Roselli et al, 2017). In vitro, CXCL12 induces an aberrant costimulatory signal after BCR activation in contrast to B cells from healthy subjects, which may lead to a form of activation-induced cell death (Roselli et al, 2017). The sustained response to vaccination seems impaired in WHIM patients, because of a humoral response that is not maintained over time (Gulino et al, 2004; Handisurya et al, 2010; Badolato et al, 2017). In this way, after a neoantigen challenge (bacteriophage immunization) of a patient, Mc Guire et al were able to demonstrate an abnormal isotype switching with a reduction in phage-specific IgG antibodies (3.3% IgG compared with 47% for controls), although the primary IgM response was normal (Mc Guire et al, 2010). Given the major role of CXCR4 in the organization of the GC, abnormalities in leukocyte trafficking in the GC may explain this humoral response defect (Allen et al, 2004; Mc Guire et al, 2010). However, data from the $Cxcr4^{+/1013}$ mouse model suggest instead a defect in the regulation of plasma cells homing in BM niches, with an aberrant accumu-lation of immature plasmablasts (see paragraph "CXCR4 in B-cell development and functions"). This remains to be demonstrated in the human disease, especially as mice do not develop hypogam-maglobulinemia, but it is an interesting avenue of research to better understand the humoral response in WHIM (Alouche et al, 2021).

The management of humoral immunodeficiency in WHIM is based on a symptomatic treatment with immunoglobulin supple-mentation, in addition to the management of neutropenia with G-CSF and antibiotic prophylaxis (Badolato et al, 2017). Adminis-tration of a CXCR4 antagonist, AMD310 or plerixafor, increases the total number of circulating B cells, but these are essentially naive B cells, and it does not improve hypogammaglobulinemia (McDermott et al, 2014, 2023). Consistently, mavorixafor, an oral CXCR4 antagonist, increases the absolute number of lymphocytes and may improve the infection rate in WHIM, but the specific effect on the humoral response is not known at present (Dale et al, 2020; Badolato & Donadieu, 2023). It is noteworthy that one spontaneous cure of WHIM syndrome has been previously reported through chromothripsis, a cellular event in which chromosomes undergo massive deletion and rearrangement. In this patient, deletions of one copy of chromosome 2, including the disease allele $CXCR4^{R334X}$, resulted in cure. Transplantation experiments in the mouse model significantly improved donor BM engraftment over BM from wild-type or WHIM syndrome model mice. This suggests that partial CXCR4 inactivation may serve as a broadly applicable strategy to HSC engraftment in transplantation (McDermott et al, 2015).

### B cell–mediated autoimmune diseases

As previously described, CXCR4 contributes to central B-cell tol-erance, thereby indicating the potential involvement of the CXCL12/CXCR4 axis in autoimmune diseases. However, few data are available on the role of CXCR4 in B cell–mediated autoimmune diseases.

In mouse models with active lupus nephritis, such as MRL/lpr or BXSB mice, a strain in which TLR7 is overexpressed, an increase in the expression of CXCR4 on the surface of B cells is observed (Wang et al, 2009). In accordance, the overexpression of CXCR4 has been demonstrated on peripheral blood B cells from active systemic lupus erythematosus (SLE) patients, especially those with lupus nephritis, a complication associated with heightened disease se-verity in flares-up (Wang et al, 2010; Hanaoka et al, 2015; Zhao et al, 2017). Concordantly, B cells from active SLE demonstrate an in-creased migration ability in a transwell-based chemotaxis assay compared with inactive patients and healthy controls (Hanaoka et al, 2015). A positive correlation between the CXCR4 level on B cells and the SLEDAI score (i.e., a disease activity score), as well as a negative correlation with the serum complement C3 level, indicative of increased consumption attributed to immune complex forma-tion and complement activation, was further described (Zhao et al, 2017).

In addition, a higher frequency of CXCR4-positive B cells is found in the interstitial lesions of lupus nephritis (Hanaoka et al, 2015; Ma et al, 2018). Using a single-cell RNA-sequencing strategy, Arazi et al recently confirmed this observation by comparing the immune cell landscape in kidney biopsies from 24 patients with lupus nephritis and 10 living donors. CXCR4 was strongly expressed in nearly all the cell clusters including all B-cell subtypes (naive B cells, activated B cells, plasma cells, and plasmablasts) (Arazi et al, 2019). Concerning CXCL12, both NZB/W mice and human SLE kidneys exhibit an increase in positive cell frequency (Balabanian et al, 2003; Wang et al, 2010). CXCL12 is concurrently expressed in the kidney epithelial cells and M2-like CD16$^+$ macrophages (Arazi et al, 2019). CXCL12 concentrations in the serum of patients remain nevertheless in the normal range (Hanaoka et al, 2015). Overall, the data obtained in both mice and humans suggest that a defect in the regulation of CXCR4 ex-pression in B cells associated with an increase in CXCL12 ex-pression in the kidney contributes to the immune infiltration of kidneys in active SLE. Of note, earlier studies reported a down-regulation of CXCR4 in B cells from SLE patients compared with healthy donors (Henneken et al, 2005; Biajoux et al, 2012). The discrepant results of these two studies may be attributed to a more heterogeneous patient population in terms of clinical manifestations, particularly lupus nephritis, and a less stringent

definition of active disease (SLEDAI score cutoff retained at 3 in Biajoux et al (2012) versus 5–10 in other studies).

More recently, an increase in TLR4⁺CXCR4⁺ double-positive circulating plasma cells has been described in active SLE patients, which also correlates with anti-dsDNA antibody titer, proteinuria, and SLEDAI (Ma et al, 2018). TLR4⁺CXCR4⁺ are terminally differentiated plasma cells, which demonstrate the capacity for producing IgG anti-dsDNA (Ma et al, 2018). Transfer of murine TLR4⁺CXCR4⁺ plasma cells into RAG-2–deficient mice induces autoantibody production and immune complex glomerulonephritis, demonstrating the role of these cells in lupus nephritis pathogenesis (Ma et al, 2018). Recent work by Chen et al seems to confirm the existence of a circulating population of mature autoantibody-producing cells expressing high levels of CXCR4 in patients with active SLE (Chen et al, 2023). These cells appear to develop pro-survival mechanisms, notably through the autocrine production of IL-10 and APRIL (Chen et al, 2023).

The mechanism explaining the increase in CXCR4 expression in active SLE has not been fully elucidated yet. JAK/STAT and PI3K/AKT pathways may be involved, as in vitro blockade of both pathways inhibits CXCR4 expression in B cells from SLE patients (Zhao et al, 2017). Zhao et al showed an inhibition of CD63 transcription, a protein that promotes trafficking of CXCR4 toward endosomes, in B cells from SLE patients compared with healthy controls. They hypothesized that the PI3K/AKT pathway in SLE contributes to CD63 down-regulation and therefore to the defect in CXCR4 internalization (Zhao et al, 2017).

Involvement of the CXCR4/CXCL12 axis is also reported in other autoimmune diseases. In active rheumatoid arthritis (RA), circulating B-cell phenotyping reveals an enhanced expression of CXCR4 compared with healthy individuals (Mahmood et al, 2020). In these patients, CXCR4⁺ terminally differentiated plasma cells accumulate in the synovial tissue, thanks to the high expression of CXCL12 in the supporting tissue (Hardt et al, 2022). In primary Sjögren's syndrome, circulating B cells also overexpress CXCR4, but Hansen et al were not able to demonstrate an enhanced migratory response to CXCL12 (Hansen et al, 2005). In bullous pemphigoid, CXCR4 expression appears increased on peripheral B cells from patients compared with controls, and a high level of CXCL12 was measured in bullous fluid, potentially attracting B cells to skin lesions. Interestingly, in vitro analysis reveals that binding of CXCL12 to CXCR4 on B cells from patients leads to plasma cell differentiation and autoantibody production in a c-Myc–dependent manner (Fang et al, 2023).

Given these results, it is essential to explore B cells in diverse autoantibody-mediated immune diseases, in order to consider targeting CXCR4.

### A new place for targeting CXCR4 in B cells?

For several years, CXCR4 has captivated drug discovery efforts (Adlere et al, 2019). However, despite the numerous data available concerning CXCR4, there remains a limited understanding of the various facets related to ligand-binding and signaling mechanisms at the molecular level, which impedes the progression of new discoveries. Recent investigations reveal that targeting of the minor allosteric subpocket yields an immunomodulatory effect (see paragraph "CXCR4 structure"), contributing to the heightened

interest in this receptor in the context of autoimmunity and inflammation (Smith et al, 2019).

So far, plerixafor or AMD3100 remains the only treatment targeting CXCR4 approved in routine use for the mobilization and collection of HSCs in autologous stem cell transplantation for multiple myeloma or lymphoma (DiPersio et al, 2009). This reversible CXCR4 antagonist is a bicyclam molecule, which binds mainly to the major binding subpocket of the receptor (Rosenkilde et al, 2007). AMD3100 and other CXCR4 antagonists were tested in oncology and HIV, but with low efficacy and high toxicity (Caspar et al, 2022). In NZB/W lupus mice, AMD3100 has been tested after short-term depletion of antibody-secreting cells with bortezomib. This treatment delays the occurrence of proteinuria and significantly improves survival, rising from 20% in the control group to 60% at 30 wk of life (Cheng et al, 2018). Likewise, the early administration of a monoclonal antibody targeting CXCL12 to NZB/W mice reduces anti-dsDNA IgG levels, postpones the initiation of proteinuria, and extends the lifespan of the animals. Indeed, at 40 wk of age, only 10% of the mice in the control group remained alive, contrasting with an 80% survival rate in the treatment group (Balabanian et al, 2003).

Recently, an unexpected immunomodulatory role of CXCR4 has emerged. Small peptides including endogenous monoamines such as histamine, clobenpropit (CB), a synthetic peptide derived from histamine, and IT1t, an isothiourea amine, inhibited pro-inflammatory cytokine production after stimulation of resiquimod, a TLR7 and TLR8 agonist (Smith et al, 2017, 2019). In vivo, daily intraperitoneal administration of IT1t over a 10-wk period in pristane-induced SLE mice was well tolerated and lowered IFN-α, TNF-α, IL-1β, and IL-17 serum levels compared with untreated mice (Smith et al, 2019). Furthermore, IT1t administration diminished anti-dsDNA production and the occurrence of lupus nephritis in mice (Smith et al, 2019). In line with these observations, PBMCs from SLE patients incubated with IT1t showed a decreased TLR7-mediated IFN-α production (Smith et al, 2019). Pro-inflammatory cytokines and the transcription of interferon-stimulated genes were also normalized in PBMCs from four juvenile SLE patients after IT1t incubation (Smith et al, 2019). Thanks to crystallographic analysis, IT1t is now known to mainly target the minor subpocket of CXCR4, which suggests a pivotal role of the minor pocket in these immunomodulatory effects (Wu et al, 2010).

Molecular mechanisms remain nevertheless to be determined. Gaining a more comprehensive understanding of the CXCR4 minor pocket's role and function is thus an intriguing research avenue, with the aim of potentially turning it into a therapeutic target in autoimmune diseases.

## Conclusion

The CXCR4 axis plays an undeniable role in orchestrating the progression of B cells throughout their developmental journey and maturation into antibody-producing cells. Precisely timed and finely regulated CXCR4 signaling, and desensitization are indispensable for maintaining B-cell homeostasis, as evidenced by the manifestation of CXCR4 GOF mutations in mice and WHIM patients.

Dysregulations within this pathway can lead to a spectrum of B cell–mediated disorders spanning from immunodeficiency to lymphoproliferative and autoimmune conditions. Although its role is not fully elucidated and requires further exploration, recent advancements in understanding the structural intricacies of CXCR4, coupled with the development of small peptides targeting the receptor's minor subpocket, hold promise for unveiling novel therapeutic avenues in this field.

## Pending issues

i Clarify the role of CXCR4 in B-cell tolerance and the retention of autoreactive B cells in the BM.
ii Further investigate the role of CXCR4 in autoimmune diseases, such as SLE, for the potential consideration of CXCR4 as a therapeutic target.
iii Decipher the function of the CXCR4 minor pocket in B cells and unravel the associated molecular mechanisms.

## Supplementary Information

## Acknowledgements

We thank the Direction de la Recherche Clinique et des Innovations (DRCI) from the Strasbourg University Hospital for supporting article processing charges. We also thank the Agence Nationale de la Recherche for its financial support (grant number ANR-21-CE15-0048 X4Inflam).

## Author Contributions

S Giorgiutti: writing—original draft, review, and editing.
J Rottura: writing—original draft, review, and editing.
A-S Korganow: writing—original draft, review, and editing.
V Gies: writing—original draft, review, and editing.

## Conflict of Interest Statement

The authors declare that they have no conflict of interest.

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
