## [Reviewer comments · Life Science Alliance]

Life Science Alliance

CXCR4: from B cell development to B-cell mediated diseases.

Stéphane Giorgiutti, Julien Rottura, Anne-Sophie Korganow and Vincent Gies

DOI: <https://doi.org/10.26508/lsa.202302465>

Corresponding author(s): Dr. Stéphane Giorgiutti (Hôpitaux Universitaires de Strasbourg; University of Strasbourg)

Review Timeline:

Submission Date:	2023-10-30
Editorial Decision:	2023-11-30
Revision Received:	2024-02-06
Editorial Decision:	2024-03-07
Revision Received:	2024-03-11
Accepted:	2024-03-12

Transaction Report:

November 30, 2023

Re: Life Science Alliance manuscript #LSA-2023-02465-T

Dr. Stéphane Giorgiutti
Hôpitaux Universitaires de STRASBOURG
Immunologie Clinique et Médecine Interne
STRASBOURG 67000
FRANCE

Dear Dr. Giorgiutti,

Thank you for submitting your manuscript entitled "CXCR4: from B cell development to B-cell mediated diseases" to Life Science Alliance. The manuscript was assessed by expert reviewers, whose comments are appended to this letter. We invite you to submit a revised manuscript addressing the Reviewer comments.

Thank you for this interesting contribution to Life Science Alliance. We are looking forward to receiving your revised manuscript.

Sincerely,

B. MANUSCRIPT ORGANIZATION AND FORMATTING:

Reviewer #1 (Comments to the Authors (Required)):

Review:

CXCR4: from B cell development to B-cell mediated diseases.

Authors:

Stéphane Giorgiutti, Julien Rottura, Anne-Sophie Korganow, Vincent Gies

The authors highlighted the importance of the chemokine receptor CXCR4 during B cell development from HSC to newly arising B cells in the bone marrow and the differentiation of plasma cells in germinal centers (GC) with a strong focus on B cell pathologies and diseases driven by impaired CXCR4 expression. Functional analyses show that loss of CXCR4 or addition of CXCL12 results in impaired HSC proliferation and maintenance. During the development of the pre-B cell antigen receptor, the B cells localize to different niches depending on the degree of maturation and dependence on Il-7 and CXCL12, where CXCR4 directs the recombination of the light chain genes, until the B cells emigrate from the bone marrow. In addition, peripheral and central tolerance is modulated by CXCR4 expression levels as the significance of the different CXCR4 levels in the light and dark zones of the GC and the T1D extrafollicular response. A gain of function (GOF) mutation impairs CXCR4 internalization and leads to e.g. Hypogammaglobulinemia, which is summarized in the WHIM syndrome. In addition, studies have revealed a link between excessive CXCR4 expression and autoimmune diseases such as SLE, which may be triggered by the CXCR4-induced Jak/STAT and PI3K/AKT pathways. A newly discovered subpocket in the CXCR4 structure could represent a new medical target.

CXCR4 is a highly conserved 7-transmembrane G-protein (GPCR), extracellularly with two negatively charged pockets, a binding site for CXCL12, intracellularly with numerous serine/threonine residues capable of signal transduction. Based on cryostructural analysis, the small pocket was identified as target of IT1t, a new CXCR4 antagonist, that inhibits the generation of pro-inflammatory cytokines and supports the role of CXCR4 as an immunomodulator.

A comprehensive collection of mainly medical data and observations as well as mouse experiments on CXCR4.

The first part focuses on the emerging phenotype in B cell development. Since there are already many very good reviews on CXCR4 chemotaxis and intracellular signaling, also other modes of action of CXCR4 are highlighted here.

In the second part, the influence of CXCR4 in coated B cell mediated diseases is examined.

Here, the focus is partly on the symptoms and their treatment with CXCR4 antagonists and the resulting therapeutic successes and further options.

Finally, the structure of CXCR4 is discussed, but only in the context of therapeutic intervention. A final short paragraph mentions the immunomodulatory properties of CXCR4 and the effect of the newly generated CXCR4 antagonists.

Introduction:

There is a general lack of contributing information,

In the beginning, it would be important to give the reader more general information about G proteins in general and CXCR4 in particular e.g. that chemokine receptors cause cells to migrate towards a cytokine gradient in what type of cells CXCR4 is expressed, and that there is interplay with other cytokine receptors.

Line 34: If HIV is mentioned directly then also clarify that CXCR4 is the co-receptor for CD4 on T cells during HIV infections etc.

B cell development and function part

The first part is comprehensive and well written.

From mature B cells to antibody response part

In this part concerning the peripheral / central tolerance it should be mentioned that IgD-BCR is essential for the functioning of CXCR4.

<https://www.pnas.org/content/early/2017/04/27/1621512114>.

How is the signaling of CXCR4 and what is the mechanism after CXCL12 binding (internalization and re-expression-homing).

In particular the reader needs information about mice: if mentioned particular genotypes there should be a short description, otherwise you can also omit the names; e.g.

Line 101: 83Igi,H-2b mice, Line 223: BXSB mice.

Line 105: The paragraph about the humanized mice seems to be too short.

Concerning the disease part:

In this part, too many observations are strung together without any explanation.

Line 184: WHIM syndrome is caused by a GOF mutation in the C-terminus of CXCR4. Why is the reader only now learning about the structure of CXCR4?

Line 191: Circulating unswitched IgD⁺ memory B cells are mentioned. Please explain the abundance of IgD BCR on memory B cells. IgD⁺ memory B cells are very rare, most of them are IgM⁺ or switched. The character of switched memory B cells is not that they are IgD⁻ but not IgM⁺ anymore.

Line 224: What characterizes SLE with lupus nephritis

Line 229: instead of serum C3 level, please state serum complement C3 level. A brief explanation would also be useful here as well.

Line 327: clarify: ...improves the survival of the animals...

Figure: the figure

The illustration is a nice overview of the different B cell stages. The absence of a legend and the absence of any reference to the individual parts in the text, the figure does not contribute to the understanding of the text.

Overall:

The advantage of this compilation is that the results of many relevant medical publications and studies on CXCR4 are mentioned. In this sense, it may help the reader to find appropriate literature for specific medical or scientific questions.

Unfortunately, there is often a lack of explanations and information that would be important for the reader to understand the research results and put them into a meaningful context.

Reviewer #2 (Comments to the Authors (Required)):

This review by Giorgiutti et al. examines B cell development, with a special focus on CXCR4 signaling in both normal and pathological conditions, particularly those involving CXCR4 gain of function mutations. The most original aspect of this review is its coverage of CXCR4's role in diseases, a topic less explored in other reviews. The review is divided into several sections:

B Cell Development Under Homeostasis: This section, already extensively covered by other reviews, should acknowledge these previous works. Key studies, particularly from the Nagasawa lab (e.g., Tokoyoda et al. *Immunity* 2004; Omatsu et al. *Immunity* 2010), are notably missing from the discussion. The review should also provide a more balanced perspective on the Clark lab's studies, which explored CXCR4 effects on Ig gene accessibility under artificial in vitro conditions. The difficulty in separating direct and indirect effects of CXCR4 signaling due to altered cell localization in vivo should be addressed.

Germinal Centers and Plasma Cell Differentiation: CXCR4's role here is primarily in facilitating the movement of centroblasts to the dark zone, rather than influencing germinal center architecture. The effects observed in WHIM patients and mouse models (e.g., Balabanian 2012, Zehentmeier *Sci Immunol* 2022) are likely due to moderate lymphopenia. Additionally, the review should correct the misconception about CXCL12 abundance in the dark zone of germinal centers.

Impact of Cxcr4 Gain of Function (GOF) in T-independent Plasma Cell Differentiation: The apparent contradiction between CXCR4 promoting quiescence in hematopoietic stem cells and B cells, versus its role in plasma cell differentiation, needs to be acknowledged. The role of plasma cell bone marrow homing and survival, both inside and outside the bone marrow environment, is still not fully understood and should be explored further, considering the impact of CXCR4 signaling deficiency on plasma cell localization in secondary lymphoid organs.

CXCR4 in Germinal Center-Derived Plasma Cells and Vaccination in WHIM Patients: This section requires a more cautious approach, as the data on vaccine responses in WHIM patients is largely anecdotal and not sufficient for definitive conclusions. The review should include and discuss Phil Murphy's lab studies, especially the significant 2015 *Cell* publication.

Sections on CXCR4 Structure and Targeting Molecules: These are well-written and appropriate.

Additional Comments:

The abstract needs to be revised for more depth and to avoid repetitive statements.

Throughout the review, there should be a greater emphasis on citing and discussing relevant studies, particularly those that have been overlooked or underrepresented.

Reviewer #3 (Comments to the Authors (Required)):

The review article by Giorgiutti et al. provides a state of the art report on CXCR4 related research focused on B-cell mediated physiological processes and disorders. The review is well written, comprehensive and well structured. The most relevant literature important to the topic are cited and discussed. The detailed description of the complex temporal and spatial expression patterns of CXCR4 and its ligand given in the text are complemented by a very good illustrative and encapsulating figure. Finally, translational aspects of novel basic research on CXCR4 structure and function relevant to clinicians are discussed.

I have no specific suggestions for improvement only some minor typing errors throughout the manuscript:

Last word of line 125: Replace "Multiples" by "Multiple"

End of line 143: Replace "increased" by "increase"

Line 189: Insert "of" between "Eighty-eight percent" and "patients"

Line 222: Replace "B-cells mediated" by "B-cell mediated"

Line 234: Replace "recently confirms" by "recently confirmed"

Last word of line 252: Replace "title" by "titre"

Line 257: Replace/rephrase "Chen et al. recent work" by "Recent work by Chen et al."

Line 265: Replace "with" by "which"

Line 279: Replace "plasma cells" by "plasma cell"

Line 295: Replace "CXCR4 gene" by "The CXCR4 gene"

Line 326: Replace "antibodies-secreting" by "antibody-secreting"

Line 341/342: Replace/rephrase "transcription of interferon-stimulated genes transcription were" by "transcription of interferon-stimulated genes were"

Line 636: Replace "characterize" by "characterizes"

Line 638: Replace "During T-dependent response" by "During the T-dependent response"

Line 645: left justify the last line

REVIEWER 1

The authors highlighted the importance of the chemokine receptor CXCR4 during B cell development from HSC to newly arising B cells in the bone marrow and the differentiation of plasma cells in germinal centers (GC) with a strong focus on B cell pathologies and diseases driven by impaired CXCR4 expression. Functional analyses show that loss of CXCR4 or addition of CXCL12 results in impaired HSC proliferation and maintenance. During the development of the pre-B cell antigen receptor, the B cells localize to different niches depending on the degree of maturation and dependence on Il-7 and CXCL12, where CXCR4 directs the recombination of the light chain genes, until the B cells emigrate from the bone marrow. In addition, peripheral and central tolerance is modulated by CXCR4 expression levels as the significance of the different CXCR4 levels in the light and dark zones of the GC and the TiD extrafollicular response. A gain of function (GOF) mutation impairs CXCR4 internalization and leads to e.g. Hypogammaglobulinemia, which is summarized in the WHIM syndrome. In addition, studies have revealed a link between excessive CXCR4 expression and autoimmune diseases such as SLE, which may be triggered by the CXCR4-induced Jak/STAT and PI3K/AKT pathways. A newly discovered subpocket in the CXCR4 structure could represent a new medical target. CXCR4 is a highly conserved 7-transmembrane G-protein (GPCR), extracellularly with two negatively charged pockets, a binding site for CXCL12, intracellularly with numerous serine/threonine residues capable of signal transduction. Based on cryostructural analysis, the small pocket was identified as target of IT1t, a new CXCR4 antagonist, that inhibits the generation of pro-inflammatory cytokines and supports the role of CXCR4 as an immunomodulator.

A comprehensive collection of mainly medical data and observations as well as mouse experiments on CXCR4. The first part focuses on the emerging phenotype in B cell development. Since there are already many very good reviews on CXCR4 chemotaxis and intracellular signaling, also other modes of action of CXCR4 are highlighted here. In the second part, the influence of CXCR4 in coated B cell mediated diseases is examined. Here, the focus is partly on the symptoms and their treatment with CXCR4 antagonists and the resulting therapeutic successes and further options. Finally, the structure of CXCR4 is discussed, but only in the context of therapeutic intervention. A final short paragraph mentions the immunomodulatory properties of CXCR4 and the effect of the newly generated CXCR4 antagonists.

Response

We thank the reviewer for her/his comments which helped us to improve the manuscript. We carefully followed and answered her/his suggestions.

INTRODUCTION**Point 1**

- There is a general lack of contributing information, In the beginning, it would be important to give the reader more general information about G proteins in general and CXCR4 in particular e.g. that chemokine receptors cause cells to migrate towards a cytokine gradient in what type of cells CXCR4 is expressed, and that there is interplay with other cytokine receptors.

Response

We have now enriched the introduction. Specifically, we provide more general information about G proteins and CXCR4, emphasizing their role in guiding cell movement in response to cytokines.

Additionally, we have restructured the review to include a dedicated section on the general aspects of CXCR4 (i.e. structure and signaling).

See lines 54 to 132 directly in the marked manuscript.

Point 2

Line 34: If HIV is mentioned directly then also clarify that CXCR4 is the co-receptor for CD4 on T cells during HIV infections etc.

Response

We now specify the role of CXCR4 as a co-receptor for HIV infection.

lines 65-67 (marked manuscript):

“In pathological contexts, CXCR4 is implicated in both cancer development and progression, as well as viral infections, particularly in the case of HIV-1, where CXCR4 serves as the co-receptor for CD4 on T cells during infection (Deng *et al*, 1996; Guo *et al*, 2016)”

B CELL DEVELOPMENT AND FUNCTION PART

Point 3

The first part is comprehensive and well written.

Response

We thank the reviewer for her/his positive comment.

FROM MATURE B CELLS TO ANTIBODY RESPONSE PART:

Point 4

- In this part concerning the peripheral / central tolerance it should be mentioned that IgD-BCR is essential for the functioning of CXCR4.

<https://www.pnas.org/content/early/2017/04/27/1621512114>.

Response

We now discuss the work of Martin Becker et al.

lines 217-224 (marked manuscript):

“*From mature B cells to antibodies response*

In periphery, CXCR4 function on mature naive B cells requires the expression of the IgD-BCR (Becker *et al*, 2017). In mice, the absence of IgD results in the abolition of CXCL12-induced migration, underlining the pivotal role of IgD in this migratory process. CXCR4 signaling further activates the ERK and AKT pathways in an IgD-dependent manner, and intriguingly, CD19 emerges as a potential intermediary between CXCR4 and IgD-BCR. This is supported by the restoration of CXCR4 signaling upon *in vitro* stimulation of CD19 in IgD-deficient B cells, indicating a functional connection between CXCR4, CD19, and IgD-BCR during B-cell activation (Becker *et al*, 2017).”

Point 5

- How is the signaling of CXCR4 and what is the mechanism after CXCL12 binding (internalization and re-expression-homing).

Response

As suggested, we have restructured the review to include a dedicated section on the general aspects of CXCR4 (i.e. structure and signaling).

Point 6

- In particular the reader needs information about mice: if mentioned particular genotypes there should be a short description, otherwise you can also omit the names; e.g. Line 101: 83Igi,H-2b mice, Line 223: BXSB mice.

Response

We have incorporated additional details to enhance the clarity of our manuscript.

- BXSB mice is a spontaneous murine model of lupus, caused by the Yaa locus, region that contain the *Tlr7* gene, on the Y chromosome. The model is therefore linked to an overexpression of TLR7. We precise this information in the manuscript:

lines 342-344 (marked manuscript):

“In mice models with active lupus nephritis, such as MRL/lpr or BXSB mice, a strain in which TLR7 is overexpressed, an increase expression of CXCR4 on the surface of B cells is observed (Wang *et al*, 2009).”

- The 3-83Igi,H-2b mb1Cre mouse is a mouse model used to study central B cell tolerance. In this strain, antibodies are directed against type I major histocompatibility complex. B cells in this strain therefore undergo central tolerance, in particular clonal deletion and receptor editing. This is now depicted in the manuscript.

lines 195-198 (marked manuscript):

“In the 3-83Igi,H-2b mb1Cre mouse model, autoreactive immature B cells, characterized by a high avidity for the self-antigen MHC-I H-2Kb, have a 1.5-fold increase in CXCR4 expression compared to non-autoreactive cells, preventing them from migrating to the periphery (Greaves *et al*, 2019; Pelanda *et al*, 2022)”

Point 7

- Line 105: The paragraph about the humanized mice seems to be too short.

Response

We expanded upon this paragraph to include a functional validation of the role of CXCR4 in retaining autoreactive cells within the bone marrow.

lines 198-206 (marked manuscript):

“This CXCR4 higher expression in immature autoreactive B cells is also observed in a human immune system humanized mouse (HIS hu-mice), in which all mouse cells express a self-antigen that reacts with developing human Igκ⁺ B cells (Alves da Costa *et al*, 2021). In this model, autoreactive immature B cells show a higher migration potential in response to CXCL12 than non-self-reactive immature B cells. Furthermore, treatment of these mice with a CXCR4 antagonist (AMD3100) results in a two-fold increase in the frequency of autoreactive B cells in the spleen after 48 hours, confirming that CXCR4 plays functionally a key role in the retention of autoreactive B cells in the BM (Alves da Costa *et al*, 2021).”

CONCERNING THE DISEASE PART

In this part, too many observations are strung together without any explanation.

We thank the reviewer for her/his remarks and developed the different points.

Point 8

Line 184: WHIM syndrome is caused by a GOF mutation in the C-terminus of CXCR4. Why is the reader only now learning about the structure of CXCR4?

Response

As previously mentioned, we have restructured the review to include a dedicated section on the general aspects of CXCR4 (i.e. structure and signaling) at the beginning of the review.

See lines 80 to 132 directly in the marked manuscript.

Point 9

Line 191: Circulating unswitched IgD⁺ memory B cells are mentioned. Please explain the abundance of IgD BCR on memory B cells. IgD⁺ memory B cells are very rare, most of them are IgM⁺ or switched. The character of switched memory B cells is not that they are IgD⁻ but not IgM⁺ anymore.

Response

The reference to circulating unswitched/switched memory B cells in WHIM syndrome in our manuscript was based on findings from two distinct studies: Anna Virginia Gulino et al (Blood, 2004) [DOI: 10.1182/blood-2003-10-3532] and Peter J. McGuire et al (Clin Immunol, 2010) [DOI: 10.1016/j.clim.2010.02.006]:

- In the Anna Virginia Gulino et al study, their phenotyping approach involved the use of CD19, CD27, and IgD markers. Although widely accepted, this panel is becoming increasingly less common in the field of cytometry. With the former, unswitched memory B cells are defined as CD27⁺IgD⁺ (utilizing a gating strategy that encompassed IgD⁺IgM⁺ B cells, as well as the infrequent IgD⁺ only memory B cells). Switched memory B cells are characterized as CD27⁺IgD⁻ (refer to Figure 5 and Table 2 in their manuscript).
- In the study by Peter J. McGuire et al, a more widely used gating strategy was employed. Switched memory B cells were defined as CD27⁺IgD⁻IgM⁻.

To ensure clarity and avoid potential confusion, as neither study made reference to the "IgD⁺ only memory B cells" subset, we have revised our manuscript.

lines 302-303 (marked manuscript):

Circulating B cells immunophenotyping show a reduction of both unswitched and switched circulating memory B cells (Gulino *et al*, 2004; Mc Guire *et al*, 2010).

Point 10

Line 224: What characterizes SLE with lupus nephritis

Response

Lupus nephritis is a potentially life-threatening manifestation of SLE. These patients are often distinguished in studies as more severe as the cutaneous and/or articular lupus and have significant disease activity in flare-ups. Furthermore, the overexpression of CXCR4 in human

SLE B cells has been described in patients with lupus nephritis, reason for focusing on these patients.

lines 344-347 (marked manuscript):

“In accordance, overexpression of CXCR4 has been demonstrated on peripheral blood B cells from active systemic lupus erythematosus (SLE) patients, especially those with lupus nephritis, a complication associated to heightened disease severity in flares-up (Wang *et al*, 2010; Zhao *et al*, 2017; Hanaoka *et al*, 2015).”

Point 11

Line 229: instead of serum C3 level, please state serum complement C3 level. A brief explanation would also be useful here as well.

Response

We have amended the text to incorporate the term "serum complement C3 level" in place of "serum C3 level," as recommended. Additionally, we have provided a brief explanation regarding the significance of serum complement C3 level in SLE.

lines 349-353 (marked manuscript):

“A positive correlation between CXCR4 level on B cells and the SLEDAI score (i.e., disease activity score), as well as a negative correlation with serum complement C3 level, indicative of increased consumption attributed to immune complex formation and complement activation, was further described (Zhao *et al*, 2017).”

Point 12

Line 327: clarify: ...improves the survival of the animals...

Response

We have now provided the specific survival percentages for each group in the Cheng *et al*. study and the Balabanian *et al* study:

- Cheng *et al*.: The survival improvement in the AMD3100-treated group, reaching 60% at 30 weeks of life, is statistically significant ($p=0.041$) compared to the control group with 20% survival.
- Balabanian *et al*.: In this study, the survival in the anti-CXCL12 monoclonal antibody group reaches 80% at 40 weeks of life, compared to 10% survival in the control group.

These pieces of information have been appropriately included in the manuscript.

lines 446-454 (marked manuscript):

“In NZB/W lupus mice, AMD3100, has been tested after short-term depletion of antibodies-secreting cells with bortezomib. This treatment delays the occurrence of proteinuria and significantly improves survival, rising from 20% in the control group to 60% at 30 weeks of life (Cheng *et al*, 2018). Likewise, the early administration of a monoclonal antibody targeting CXCL12 to NZB/W mice reduces anti-dsDNA IgG levels, postpones the initiation of proteinuria, and extends the lifespan of the animals. Indeed, at 40 weeks of age, only 10% of the mice in the control group remained alive, contrasting with an 80% survival rate in the treatment group (Balabanian *et al*, 2003).

THE FIGURE

Point 13

The illustration is a nice overview of the different B cell stages. The absence of a legend and the absence of any reference to the individual parts in the text, the figure does not contribute to the understanding of the text.

Response

We apologize for the lack of clarity. The legend was placed alone at the end of the manuscript after the references. Please find below the revised legend of the figure. In our revision, as required by the editorial guidelines, we submit the figure in a distinct file. Additionally, as suggested, we have incorporated multiple references to the figure in the text.

lines 807-824 (marked manuscript):

“Figure 1. CXCR4 and B cell development. CXCR4 is involved in maintaining the HSC pool throughout contacts with CXCL12-abundant reticular (CAR) cells in the bone marrow. The level of CXCR4 expression varies during B cell differentiation. Pro-B cells undergo intensive proliferation in the vicinity of IL-7⁺ CXCL12⁺ stromal cells. An upregulation of CXCR4 expression characterizes pre-B cells stage, with a disengagement from IL7-rich niches. Then, low level of CXCR4 expression in immature B cells induces bone marrow egress to secondary lymphoid organs (see paragraph “*Central development: from hematopoietic stem cells to immature B cells*”). During the T-dependent response in germinal center, centroblasts in the dark zone, abundant in CXCL12-expressing stromal cells, strongly express CXCR4, undergo proliferation and somatic hypermutation. Centrocytes downregulate CXCR4 to enter in the light zone through the CXCL13/CXCR5 axis. Plasma cells emerging from the germinal center re-express CXCR4 and accumulate in the bone marrow, where some of them mature into long-lived plasma cells. Regulation of CXCR4 signaling is also required to limit extrafollicular response and avoid plasmablasts accumulation in the bone marrow (see paragraph “*From mature B cells to antibodies response*”). BCR: B cell receptor; CAR: CXCL12 abundant reticular cells; CLP: common lymphoid progenitor; HSC: hematopoietic stem cells; SHM: somatic hypermutation. The figure was partly generated using Servier Medical Art, provided by Servier, licensed under a Creative Commons Attribution 3.0 unported license.”

OVERALL

The advantage of this compilation is that the results of many relevant medical publications and studies on CXCR4 are mentioned. In this sense, it may help the reader to find appropriate literature for specific medical or scientific questions. Unfortunately, there is often a lack of explanations and information that would be important for the reader to understand the research results and put them into a meaningful context.

Response

We have incorporated the different points into our manuscript, especially addressing the identified lack of explanations and contextual information. We believe that these modifications enhance the overall quality and thank the reviewer for her/his help.

REVIEWER 2

This review by Giorgiutti et al. examines B cell development, with a special focus on CXCR4 signaling in both normal and pathological conditions, particularly those involving CXCR4 gain of function mutations. The most original aspect of this review is its coverage of CXCR4's role in diseases, a topic less explored in other reviews. The review is divided into several sections:

Response

We thank the reviewer for her/his comments. We carefully followed and addressed the suggestions, completing our work as depicted below.

B CELL DEVELOPMENT UNDER HOMEOSTASIS

Point 1

This section, already extensively covered by other reviews, should acknowledge these previous works. Key studies, particularly from the Nagasawa lab (e.g., Tokoyoda et al. *Immunity* 2004; Omatsu et al. *Immunity* 2010), are notably missing from the discussion.

Response

- The Tokoyoda et al. article (*Immunity*, 2004, [10.1016/j.immuni.2004.05.001](https://doi.org/10.1016/j.immuni.2004.05.001)) is referenced in the 'peripheral' section of the review. We are now incorporating and further analyzing this study in the 'central' part, as it notably complements the data presented by Cordeiro Gomes et al. (*Immunity*, 2016).
- The study of Omatsu et al. (*Immunity*, 2010; [10.1016/j.immuni.2010.08.017](https://doi.org/10.1016/j.immuni.2010.08.017)), describing the selective short-term ablation of CXCL12-abundant reticular (CAR) cells, underscore and strengthen the correlation between CAR cells, the quantity of hematopoietic stem cells (HSCs), and the development of lymphoid and erythroid progenitors, along with the crucial role in maintaining HSCs in an undifferentiated state. We have now incorporated this study alongside the findings of Sugiyama et al. (*Immunity*, 2006; [10.1016/j.immuni.2006.10.016](https://doi.org/10.1016/j.immuni.2006.10.016)). The combined insights from these works, both from the Nagasawa lab, contribute to a more comprehensive understanding of the intricate interplay within the CXCL12/CXCR4 axis and its impact on hematopoiesis.

Lines 150-164 (marked manuscript):

“Depletion of CAR cells in mice results in a reduced number of HSCs and an upregulation of early myeloid selector genes, resembling the phenotype observed in wild-type HSCs cultured without a niche (Omatsu *et al*, 2010). Conversely, aberrant CXCR4 signaling is also deleterious to generate early lymphoid progenitors, as the increased quiescence of short-term HSCs observed in CXCR4 GOF *Cxcr4*^{+/-1013} mice impairs transition to multipotent progenitors (MPP) and to the common lymphoid progenitor (CLP) (Freitas *et al*, 2017). Hence, both CAR cells and a fine regulation of CXCR4 signaling are essential for the generation of lymphoid progenitors, as well as for the maintenance of HSCs in an undifferentiated state (Omatsu *et al*, 2010).

Furthermore, CXCR4 enables CLP to position in the vicinity of IL7⁺CXCL12⁺ stromal cells, allowing the commitment of CLP in the B lineage (Cordeiro Gomes *et al*, 2016; Kaiser *et al*, 2023; Tokoyoda *et al*, 2004). Subsequently, pro-B cells in contact with the IL-7-expressing cells proliferate, while remaining anchored within these niches (Tokoyoda *et al*, 2004)”

Point 2

The review should also provide a more balanced perspective on the Clark lab's studies, which explored CXCR4 effects on Ig gene accessibility under artificial *in vitro* conditions. The difficulty in separating direct and indirect effects of CXCR4 signaling due to altered cell localization *in vivo* should be addressed.

Response

In addressing the reviewer's point regarding the Clark lab's studies, we have taken steps to provide a more balanced perspective. We now underline the complexity of discerning between direct and indirect effects of CXCR4 signaling, especially considering the altered cell localization *in vivo*. We have incorporated explicit references to the artificial culture conditions utilized in the Clark lab's studies.

Lines 180-182 (marked manuscript):

“Discerning between the direct and indirect effects of CXCR4 signaling *in vivo* poses a considerable challenge. Numerous data from studies conducted in artificial bone marrow progenitor culture systems indicates that CXCR4 exerts effects beyond mere chemotaxis. “

Lines 207-209 (marked manuscript):

“More recently, Okoreeh *et al.* suggested, with *in vitro* culture of B220⁺IgM⁺ progenitors, the direct requirement for CXCR4 signaling in orchestrating receptor editing and Igλ recombination (Okoreeh *et al.*, 2022).”

GERMINAL CENTERS AND PLASMA CELL DIFFERENTIATION:

Point 3

CXCR4's role here is primarily in facilitating the movement of centroblasts to the dark zone, rather than influencing germinal center architecture. The effects observed in WHIM patients and mouse models (e.g., Balabanian 2012, Zehentmeier *Sci Immunol* 2022) are likely due to moderate lymphopenia. Additionally, the review should correct the misconception about CXCL12 abundance in the dark zone of germinal centers.

Response

We are now more balanced with the notion of GC "architecture" and have refined our statement on CXCL12-expressing reticular cells to avoid any confusion.

Furthermore, given the current paucity of evidence, one can indeed only make assumptions regarding the origin of abnormal GC structure in WHIM patients. In response to this uncertainty, we have tempered our statement and introduced the notion of lymphopenia as a potential contributing factor.

Lines 225-232 (marked manuscript):

“Furthermore, CXCR4 is essential for the T-dependent immune response notably through its role in the mobilization of centroblasts in the germinal center (GC) of secondary lymphoid organs. Centroblasts located in the dark zone of the GC, which contains a network of CXCL12-expressing reticular cells, exhibit strong CXCR4 expression in both mice and humans (Allen *et al.*, 2004; Caron *et al.*, 2009; Rodda *et al.*, 2015). In contrast, centrocytes downregulate CXCR4, enabling them to leave the dark zone and to enter in the light zone

directed by the CXCL13/CXCR5 axis (Allen *et al*, 2004; Caron *et al*, 2009; Allen *et al*, 2007; Victora *et al*, 2010).”

Lines 241-246 (marked manuscript):

“As a result, altered CXCR4 expression could have an impact on the homeostasis of the GC. In accordance with this, altered lymphoid follicle architecture has been reported in CXCR4 GOF *Cxcr4*⁺¹⁰¹³ mice and in two histopathological reports of lymph nodes from WHIM patients, but the origin of these abnormalities is difficult to interpret in the context of WHIM-associated lymphopenia (Balabanian *et al*, 2012; Mentzer *et al*, 1977; Zuelzer, 1964).”

Point 4

Impact of Cxcr4 Gain of Function (GOF) in T-independent Plasma Cell Differentiation:

The apparent contradiction between CXCR4 promoting quiescence in hematopoietic stem cells and B cells, versus its role in plasma cell differentiation, needs to be acknowledged.

Response

This notion is now introduced in the manuscript.

Lines 282-284 (marked manuscript):

“Altogether, fine regulation of CXCR4 signaling is necessary to obtain a specific and durable humoral response. The apparent contradiction between its roles in plasma cell differentiation and HSC quiescence underscores CXCR4's context-dependent functions across cell types.”

Point 5

The role of plasma cell bone marrow homing and survival, both inside and outside the bone marrow environment, is still not fully understood and should be explored further, considering the impact of CXCR4 signaling deficiency on plasma cell localization in secondary lymphoid organs.

Response

The need for further exploration has been incorporated both into the “From mature B cells to antibodies response” part and into the conclusion.

Lines 274-275 (marked manuscript):

“This highlights our incomplete understanding of plasma cell homing and survival, both within and outside the bone marrow environment.”

Lines 476-485 (marked manuscript):

“The CXCR4 axis plays an undeniable role in orchestrating the progression of B cells throughout their developmental journey and maturation into antibody-producing cells. Precisely timed and finely regulated CXCR4 signaling, and desensitization are indispensable for maintaining B cell homeostasis, as evidenced by the manifestation of *CXCR4* GOF mutations in mice and WHIM patients. Dysregulations within this pathway can lead to a spectrum of B-cell mediated disorders spanning from immunodeficiency to lymphoproliferative and autoimmune conditions. Although its role is not fully elucidated and requires further exploration, recent advancements in understanding the structural intricacies of CXCR4, coupled with the development of small peptides targeting the receptor's minor subpocket, hold promise for unveiling novel therapeutic avenues in this field.”

CXCR4 IN GERMINAL CENTER-DERIVED PLASMA CELLS AND VACCINATION IN WHIM PATIENTS

Point 6

This section requires a more cautious approach, as the data on vaccine responses in WHIM patients is largely anecdotal and not sufficient for definitive conclusions.

Response

We acknowledge that the data on vaccine response in WHIM syndrome are based on only a few clinical cases. We have therefore exercised caution in the manuscript.

Lines 279-281 (marked manuscript):

“This is consistent with a defect of durable humoral T-dependent immune response (i.e., after vaccination) observed in some WHIM patients (Gulino *et al*, 2004; Handisurya *et al*, 2010).”

Lines 308-310 (marked manuscript):

“The sustained response to vaccination seems impaired in WHIM patients, due to a humoral response that is not maintained over time (Gulino *et al*, 2004; Handisurya *et al*, 2010; Badolato *et al*, 2017).”

Point 7

The review should include and discuss Phil Murphy's lab studies, especially the significant 2015 Cell publication.

Response

Our review already includes two clinical trials conducted by PM Murphy, and the noteworthy 2015 Cell publication ([10.1016/j.cell.2015.01.014](https://doi.org/10.1016/j.cell.2015.01.014)) details a case of spontaneous WHIM cure through chromothripsis in HSCs. This event resulted in the deletion of 163 genes on chromosome 2, including the gain-of-function mutant CXCR4.

While we acknowledge the significance of this work, our review maintains a primary focus on B cells, which informed our decision not to incorporate it initially. To prevent diverting attention from our primary focus, we have integrated the findings from this study at the end of the WHIM section within the clinical management part.

Lines 330-336 (marked manuscript):

“Noteworthy, one spontaneous cure of WHIM syndrome has been previously reported through chromothripsis, a cellular event in which chromosomes undergo massive deletion and rearrangement. In this patient, deletions of one copy of chromosome 2, including the disease allele $CXCR4^{R334X}$, resulted in cure. Transplantation experiments in mice model, significantly improved donor BM engraftment over BM from wild-type or WHIM syndrome model mice. This suggests that partial CXCR4 inactivation may serve as a broadly applicable strategy to HSC engraftment in transplantation (McDermott *et al*, 2015).”

SECTIONS ON CXCR4 STRUCTURE AND TARGETING MOLECULES

Point 8

These are well-written and appropriate.

Response

We thank the reviewer for her/his positive comment.

ADDITIONAL COMMENTS

The abstract needs to be revised for more depth and to avoid repetitive statements. Throughout the review, there should be a greater emphasis on citing and discussing relevant studies, particularly those that have been overlooked or underrepresented.

Response

We carefully reviewed our manuscript in light of the reviewer's comments. The manuscript has undergone significant improvements. We have enhanced and expanded the abstract, ensuring it adheres to the 175-word limit.

Lines 27-39 (marked manuscript):

“Chemokines receptors are members of the G protein-coupled receptor superfamily. The C-X-C chemokine receptor type 4 (CXCR4), one of the most studied chemokine receptors, is widely expressed in hematopoietic and immune cell populations. It is involved in leukocyte trafficking in lymphoid organs and inflammatory sites through its interaction with its natural ligand CXCL12. CXCR4 assumes a pivotal role in B cell development, ranging from early progenitors to the differentiation of antibody-secreting cells. This review emphasizes the significance of CXCR4 across the various stages of B cell development, including central tolerance, and delves into the association between CXCR4 and B-cell-mediated disorders, from immunodeficiencies such WHIM (warts, hypogammaglobulinemia, infections, and myelokathexis) syndrome to autoimmune diseases like systemic lupus erythematosus. The potential of CXCR4 as a therapeutic target is discussed, especially through the identification of novel molecules capable of modulating specific pockets of the CXCR4 molecule. These insights provide a basis for innovative therapeutic approaches in the field.”

REVIEWER 3

The review article by Giorgiutti et al. provides a state-of-the-art report on CXCR4 related research focused on B-cell mediated physiological processes and disorders. The review is well written, comprehensive and well structured. The most relevant literature important to the topic are cited and discussed. The detailed description of the complex temporal and spatial expression patterns of CXCR4 and its ligand given in the text are complemented by a very good illustrative and encapsulating figure. Finally, translational aspects of novel basic research on CXCR4 structure and function relevant to clinicians are discussed.

I have no specific suggestions for improvement only some minor typing errors throughout the manuscript:

- Last word of line 125: Replace "Multiples" by "Multiple"
- End of line 143: Replace "increased" by "increase"
- Line 189: Insert "of" between "Eighty-eight percent" and "patients"
- Line 222: Replace "B-cells mediated" by "B-cell mediated"
- Line 234: Replace "recently confirms" by "recently confirmed"
- Last word of line 252: Replace "title" by "titre"
- Line 257: Replace/rephrase "Chen et al. recent work" by "Recent work by Chen et al."
- Line 265: Replace "with" by "which"
- Line 279: Replace "plasma cells" by "plasma cell"
- Line 295: Replace "CXCR4 gene" by "The CXCR4 gene"
- Line 326: Replace "antibodies-secreting" by "antibody-secreting"
- Line 341/342: Replace/rephrase "transcription of interferon-stimulated genes transcription were" by "transcription of interferon-stimulated genes were"
- Line 636: Replace "characterize" by "characterizes"
- Line 638: Replace "During T-dependent response" by "During the T-dependent response"
- Line 645: left justify the last line

Response

We thank the reviewer for her/his positive comments. We have addressed and rectified all the highlighted points/typos.

March 7, 2024

RE: Life Science Alliance Manuscript #LSA-2023-02465-TR

Dr. Stéphane Giorgiutti
Hôpitaux Universitaires de Strasbourg
Immunologie Clinique et Médecine Interne
1 place de l'hôpital
STRASBOURG 67000
France

Dear Dr. Giorgiutti,

Thank you for submitting your revised manuscript entitled "CXCR4: from B cell development to B-cell mediated diseases.". We would be happy to publish your paper in Life Science Alliance pending final revisions necessary to meet our formatting guidelines.

- please be sure that the authorship listing and order is correct
- please add the Twitter handle of your host institute/organization as well as your own or/and one of the authors in our system
- please add an Author Contributions section to your main manuscript text

A. FINAL FILES:

B. MANUSCRIPT ORGANIZATION AND FORMATTING:

Sincerely,

Reviewer #1 (Comments to the Authors (Required)):

The manuscript has been significantly improved. All points of discussion have been addressed and changed in the manuscript. I would recommend publication.

Reviewer #2 (Comments to the Authors (Required)):

The authors addressed all of questions and concerns raised and I believe this manuscript is now suitable for publication.

March 12, 2024

RE: Life Science Alliance Manuscript #LSA-2023-02465-TRR

Dr. Stéphane Giorgiutti
Hôpitaux Universitaires de Strasbourg
Immunologie Clinique et Médecine Interne
1 place de l'hôpital
STRASBOURG 67000
France

Dear Dr. Giorgiutti,

Thank you for submitting your Review entitled "CXCR4: from B cell development to B-cell mediated diseases.". It is a pleasure to let you know that your manuscript is now accepted for publication in Life Science Alliance. Congratulations on this interesting work.

DISTRIBUTION OF MATERIALS:

Again, congratulations on a very nice paper. I hope you found the review process to be constructive and are pleased with how the manuscript was handled editorially. We look forward to future exciting submissions from your lab.

Sincerely,
